# ECONOMICAL HYPERPARAMETER OPTIMIZATION WITH BLENDED SEARCH STRATEGY

**Chi Wang**[*]**, Qingyun Wu**[*]**, Silu Huang, Amin Saied**
Microsoft Corporation, Redmond, WA 98052, USA
{wang.chi,qingyun.wu,silu.huang,amin.saied}@microsoft.com

## ABSTRACT

We study the problem of using low cost to search for hyperparameter configurations in a large search space with heterogeneous evaluation cost and model quality. We propose a blended search strategy to combine the strengths of global and local search, and prioritize them on the fly with the goal of minimizing the total cost spent in finding good configurations. Our approach demonstrates robust performance for tuning both tree-based models and deep neural networks on a large AutoML benchmark, as well as superior performance in model quality, time, and resource consumption for a production transformer-based NLP model fine-tuning task.

## 1 INTRODUCTION

Hyperparameter optimization (HPO) of modern machine learning models is a resource-consuming task, which is unaffordable to individuals or organizations with little resource (Yang & Shami, 2020). Operating HPO in a low-cost regime has numerous benefits, such as democratizing ML techniques, enabling new applications of ML, which requires frequent low-latency tuning, and reducing the carbon footprint. It is inherently challenging due to the nature of the task: trying a large number of configurations of *heterogeneous* cost and accuracy in a large search space. The expense can accumulate from multiple sources: either a large number of individually cheap trials or a small number of expensive trials can add up the required resources.

There have been multiple attempts to address the efficiency of HPO from different perspectives. Each of them has strengths and limitations. For example, Bayesian optimization (BO) (Brochu et al., 2010), which is a class of global optimization algorithms, is used to minimize the total number of iterations to reach global optima. However, when the cost of different hyperparameter configurations is heterogeneous, vanilla BO may select a configuration that incurs unnecessarily high cost. As opposed to BO, local search (LS) methods (Wu et al., 2021) are able to control total cost by preventing very expensive trials until necessary, but they may get trapped in local optima. Multi-fidelity methods (Jamieson & Talwalkar, 2016) aim to use cheap proxies to replace some of the expensive trials and approximate the accuracy assessment, but can only be used when such proxies exist. A single search strategy is difficult to meet the generic goal of economical HPO.

In this work, we propose a blended search strategy which combines global search and local search strategy such that we can enjoy benefits from both worlds: (1) global search can ensure the convergence to the global optima when the budget is sufficient; and (2) local search methods enable a better control on the cost incurred along the search trajectory. Given a particular global and local search method, our framework, which is named as `BlendSearch`, combines them according to the following design principles. (1) Instead of sticking with a particular method for configuration selection, we consider both of the candidate search methods and decide which one to use at each round of the configuration selection. (2) We use the global search method to help decide the starting points of local search threads. (3) We use the local search method to intervene the global search method's configuration selection to avoid configurations that may incur unnecessarily large evaluation cost. (4) We prioritize search instances of both methods according to their performance and efficiency of performance improvement on the fly. Extensive empirical evaluation on the AutoML

---

[*]Equal contribution

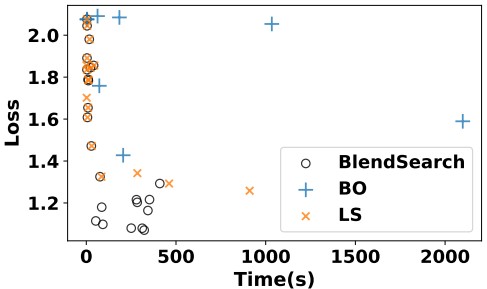
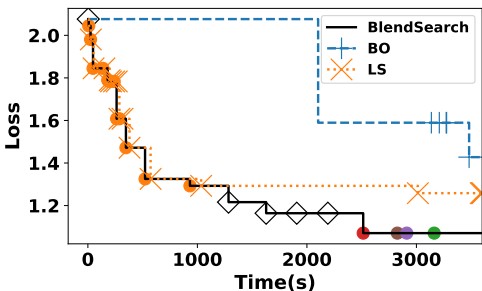

(a) Loss vs. evaluation time for configs tried by each method. One point represents one config. The lower is the loss the better is the quality of the config. Longer evaluation time corresponds to larger training cost

(b) Best loss vs. optimization time per method. Colored circles – configs proposed by LS threads in our method (color change indicates thread change); diamonds – configs proposed by BO in our method

Figure 1: A typical example of the different behaviors of BO, LS and our proposed `BlendSearch` in tuning a set of 11-dim hyperparameters for XGBoost. BO is prone to selecting expensive but not necessarily good configs. LS avoids expensive configs in the beginning but is prone to getting stuck in local regions. `BlendSearch` switches between one BO and multiple LS search threads and prioritizes the more promising ones, and turns out to try more low-cost, high-quality configs.

Benchmark (Gijsbers et al., 2019) validates the robust performance of our method on a wide variety of datasets. `BlendSearch` is now publicly available in an open-source AutoML Library[1].

## 2 BACKGROUND AND RELATED WORK

We first briefly introduce the vanilla Bayesian optimization methods and local search methods, which are among the building blocks of our method. Bayesian optimization is a class of global optimization algorithms which is suitable for optimizing expensive black-box functions. It models the probabilistic distribution of the objective conditioned on the optimization variables. Typical models include Gaussian process (Snoek et al., 2012), random forest (Hutter et al., 2011), and tree Parzen estimator (TPE) (Bergstra et al., 2011). In BO methods, an acquisition function is used to determine the next point to evaluate. Two common acquisition functions are the expected improvement (EI) (Bull, 2011) over the currently best-observed objective and upper confidence bound (UCB) (Srinivas et al., 2009). Local search methods are prevalent in the general optimization literature (Spall et al., 1992; Nesterov & Spokoiny, 2017) but less studied in the HPO literature due to the possibility of getting trapped in local optima (György & Kocsis, 2011). Recent work (Wu et al., 2021) shows that a local search method $FLOW^2$ can make HPO cost-effective when combined with low-cost initialization and random restart. At each iteration, it samples a pair of vectors (with opposite directions) uniformly at random from a unit sphere, the center of which is the best configuration found so far (a.k.a. incumbent) and the radius of which is the current stepsize. Expensive configurations are avoided in the beginning as each iteration proposes a configuration near the incumbent. Random restart of the local search is performed once the convergence condition is satisfied.

There are several attempts to address the limitations of vanilla BO or local search methods. BOwLS (BO with local search) (Gao et al., 2020) uses a BO model to select the starting point of a local search thread. Each local search thread is run until convergence and the BO model is updated with the start point and the converged loss. Trust region BO (Eriksson et al., 2019) fits a fixed number of local models and performs a principled global allocation of samples across these models via an implicit bandit approach. It is primarily designed for HPO problems with high-dimensional numerical hyperparamters. Unfortunately, all existing work that tries to combine global search with local search methods does not consider the heterogeneity of evaluation cost incurred along with the search. There are also a lot of attempts in making HPO efficient by speeding up configuration evaluation. Multi-fidelity optimizations (Klein et al., 2017; Li et al., 2017; Kandasamy et al., 2017; Falkner et al., 2018; Lu et al., 2019; Li et al., 2020) are proposed for this purpose. They usually require an additional degree of freedom in the problem called 'fidelity', to allow performance assessment on a configura-

---

[1]`https://github.com/microsoft/FLAML`

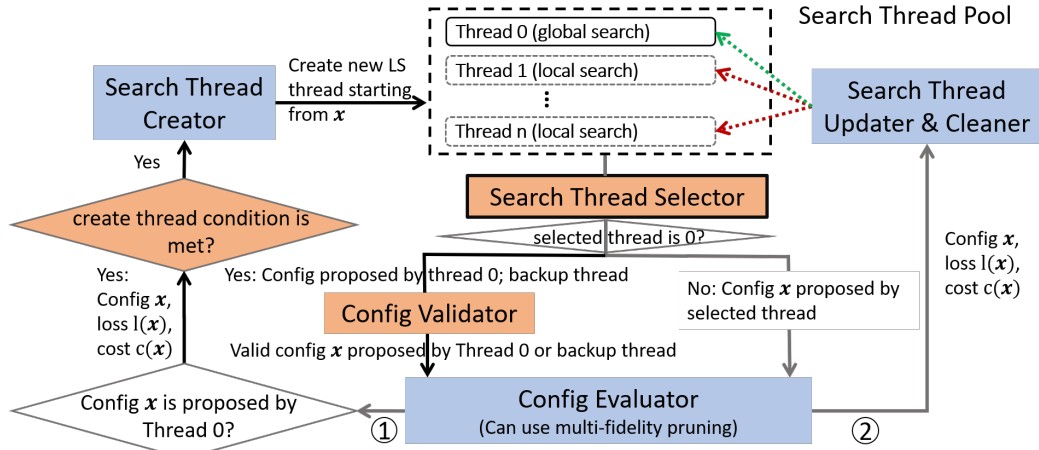

Figure 2: Framework. Both paths from 'Config Evaluator' are executed independently.

tion with different fidelities. There is surprisingly little prior work for generic cost-effective HPO. Gaussian process with expected improvement per second (GPEIPS) (Snoek et al., 2012) models the evaluation cost using another Gaussian process, and heuristically adds the estimated cost into the acquisition function. It does not always outperform GPEI as the acqusition function can overpenalize good but expensive configurations.

## 3   BLENDSEARCH

Our framework needs the following information as inputs.

- $B$ is the total budget of cost. In this work, we measure the cost by CPU/GPU time.
- $P$ is the input HPO problem, which has the following attributes characterizing the problem.
  - $P.\mathcal{X}$ is the search space in which each element $\mathbf{x}$ is a $d$-dimensional hyperparameter configuration. For a non-categorical hyperparameter coordinate $i \in [d]$, if different values of $\mathbf{x}_i$ lead to heterogeneous cost and there is a known value $P.x_i^{LowCost}$ corresponding to a low cost, it is considered as a controlled dimension. We use $P.D$ to denote such a set of controlled dimensions.
  - $P.\texttt{LossFunc}(\cdot)$ is the loss function to be minimized in terms of the configurations $\mathbf{x} \in P.\mathcal{X}$.
  - $P.\texttt{CostFunc}(\cdot)$ is the cost function that outputs the cost incurred when evaluating $\mathbf{x}$.
  
  The goal of an HPO algorithm is to minimize the loss $P.\texttt{LossFunc}(\mathbf{x})$ with the constraint that the total cost incurred $G(\pi) \coloneqq \sum_{\mathbf{x} \in \mathcal{I}(\pi)} P.\texttt{CostFunc}(\mathbf{x}) \leq B$, where $\mathcal{I}(\pi)$ is the search trajectory of algorithm $\pi$. Note that both $P.\texttt{LossFunc}(\mathbf{x})$ and $P.\texttt{CostFunc}(\mathbf{x})$ are black-box functions meaning that typically the analytic form is not available, and only function values can be observed. In order to distinguish the operation of querying from loss/cost function and the loss/cost observation, we use $P.\texttt{LossFunc}(\mathbf{x})$ and $P.\texttt{CostFunc}(\mathbf{x})$ to denote the former, and use $l(\mathbf{x})$ and $c(\mathbf{x})$ to denote the latter. $l$ and $c$ (omitting $\mathbf{x}$) are used when there is no ambiguity.
- $\mathcal{G}$ is the global search method to be used. $\mathcal{L}$ is the local search method to be used. $\mathcal{L}.\Delta$ is the largest stepsize used in local search method $\mathcal{L}$, i.e., the largest possible change on a hyperparameter value between two consecutive search steps.

The overall design of our framework is presented in Figure 2 and Algorithm 1. The key idea is to maintain a pool of search threads, one of which corresponds to global search and the others local search. The pool starts with one global search thread and gradually adds local search threads as the search goes on. Here a search thread is an instance of a global search or local search method, each with its own search trajectory. At each round, a *search thread selector* selects one of the search threads from the pool according to a priority metric that reflects the search threads' current performance and efficiency of performance improvement. The selected search thread will then be used to propose a configuration to evaluate in this round. When the selected search thread is the global search thread, a *config validator* first checks whether the proposed configuration is within

the 'admissible' region for evaluation. If not, it uses a backup local search thread instead. A local search thread will be created only when the global search thread proposes a valid config and a certain *create thread condition* is met, and will be deleted once it converges. Two local search threads will be merged into one if they are close enough. The priority of each search thread is updated after each evaluation round.

---

**Algorithm 1** `BlendSearch`

---

    **Inputs:** HPO problem $P$, a global search method $\mathcal{G}$, a local search method $\mathcal{L}$, total budget $B$.
1: **Initialization:** Initialize $\mathcal{F}.\mathbb{S} = [S_0]$ where $S_0$ is an instance of $\mathcal{G}$. We denote by $\mathcal{F}.\mathbf{x}_0$ the initial point of the search. By design, the values of the controlled dimensions of $\mathcal{F}.\mathbf{x}_0$ are set to be $P.\mathbf{x}^{LowCost}$ and those for the other dimensions are proposed by $S_0$.
2: **while** $\mathcal{F}.c < B$ **do**
3:     $\tilde{S}, \tilde{S}_{\text{bak}} \leftarrow$ `SelectThread`$(\mathcal{F})$
4:     $(\mathbf{x}, l, c) \leftarrow$ `SearchEvaluate1Step`$(\tilde{S}, \mathcal{F}, P)$
5:     **if** $\mathbf{x}$ is `invalid` **then** $(\mathbf{x}, l, c) \leftarrow$ `SearchEvaluate1Step`$(\tilde{S}_{\text{bak}}, \mathcal{F}, P)$
6:     **if** $\mathbf{x}$ is proposed by global search & `CreateNewLSCondition` is satisfied **then**
7:         Initialize $S =$ `InitializeSearchThread`$(\mathcal{L}, P, (\mathbf{x}, l, c))$
8:         Add the new LS thread $S$ into the pool: $\mathcal{F}.\mathbb{S} \leftarrow \mathcal{F}.\mathbb{S} + S$
9:     `DeleteAndMergeLS`$(\mathcal{F})$         ▷ Merge or delete existing LS threads when necessary
10:     Update $\mathcal{F}.$`Priority`
11:     **if** $\mathbf{x}$ is proposed by global search **then** `UpdateGSModel`$(S_0, (\mathbf{x}, l, c))$

---

For convenience, we use $\mathcal{F}$ to denote a collection of framework-level variables:

- $\mathcal{F}.\mathbb{S}$ is the list of search threads maintained in our framework. $\mathcal{F}.\mathbb{S}$ contains at least one search thread and among them there is one and only one global search thread, i.e., $S_0$, in $\mathcal{F}.\mathbb{S}$.
- $\mathcal{F}.$`Priority` is a priority dictionary, in which the keys are the search threads in $\mathcal{F}.\mathbb{S}$, and the values are the priority of the corresponding search threads.
- Bookkeeping information of $\mathcal{F}$: $\mathcal{F}.l^*$ is the best loss achieved among all the search threads and $\mathcal{F}.c$ is the total cost consumed in our framework.
- $\mathcal{F}.\mathcal{R}$ is the 'admissible' region on the controlled dimensions of the current search, which is a hyperrectangle and can be written in the form of $\mathcal{F}.\mathcal{R} := \{[\mathcal{F}.x_i^{\min}, \mathcal{F}.x_i^{\max}]\}_{i \in D}$ with $\mathcal{F}.x_i^{\min}$ and $\mathcal{F}.x_i^{\max}$ denoting the minimum and maximum value along the $i$-th dimension in $\mathcal{F}.\mathcal{R}$ respectively. They are initially set as $\mathcal{F}.x_i^{\min} = \mathcal{F}.x_i^{\max} = P.x_i^{LowCost}$ for all $i \in D$. The 'admissible' region gradually expands during the search: (1) it is expanded to cover all the points evaluated by all the search threads and all the points that are possible to be reached by the local search within one search step, as shown in line 7 and 8 of Algorithm 2; (2) it expands if a local search thread converges, as shown in line 3 of Algorithm 7 (included in Appendix A).

In the following, we explain the key steps in our algorithm.

**Step 1: Search thread selector (line 3 of Alg 1).** In addition to the primary search thread $\tilde{S}$, `SelectThread` also outputs a backup search thread $\tilde{S}_{\text{bak}}$ which is guaranteed to be a local search thread. It is set to be none when there is no local search thread yet in $\mathcal{F}.\mathbb{S}$. Specifically,

$$\tilde{S} = \arg\max_{S \in \mathcal{F}.\mathbb{S}} \mathcal{F}.\text{Priority}(S), \tilde{S}_{\text{bak}} = \begin{cases} \arg\max_{S \in (\mathcal{F}.\mathbb{S} \setminus S_0)} \mathcal{F}.\text{Priority}(S) & \mathcal{F}.\mathbb{S} \setminus S_0 \neq \emptyset \\ \text{None} & \mathcal{F}.\mathbb{S} \setminus S_0 = \emptyset, \end{cases}$$
(1)

The design of the priority metric follows the principle of optimism in the face of uncertainty from the multi-armed bandit problem to balance exploitation and exploration (Lattimore & Szepesvári, 2020). Specifically, linear extrapolation is performed adaptively and locally to calculate the improvement speed of each search thread. The estimated future reward based on such a linear extrapolation provides a first-order upper bound of the ground truth future reward assuming each search thread has a diminishing return, i.e., the speed of improvement decreases as more resource is spent. Formally, we introduce the following variables and functions for each search thread $S \in \mathcal{F}.\mathbb{S}$.

- Bookkeeping information: $S.l^{1\text{st}}$ and $S.l^{2\text{nd}}$ are the best loss so far and second best loss before the best loss is achieved. $S.c^{1\text{st}}$ and $S.c^{2\text{nd}}$ are the total cost taken when $S.l^{1\text{st}}$ and $S.l^{2\text{nd}}$ are achieved respectively. $S.c$ is the total cost spent in $S$. $S.\mathbf{x}^{1\text{st}}$ is the best configuration found so far.

- $S.s$ is the performance improvement speed of $S$. It is calculated as $S.s = \frac{S.l^{2\mathrm{nd}} - S.l^{1\mathrm{st}}}{S.c - S.c^{2\mathrm{nd}}}$. This formula is only valid when there is at least one improvement. Otherwise, we do not have enough information to estimate the speed of improvement. We set the speed to the highest speed of all the search threads when $S.l^{2\mathrm{nd}} = S.l^{1\mathrm{st}}$. It is due to an implicit assumption of diminishing return.
- $S.x^{\min}$ and $S.x^{\max}$ are the minimum and maximum value of the $i$-th dimension of all hyperparameters configurations evaluated in $S$ respectively.
- $S.\texttt{CostImp}(\cdot)$ is a function whose input is a target loss and output is the anticipated cost for $S$ to generate a better loss than this target. We use the following formula to compute it, which is intuitively using the cost for improvement in the past to estimate that in the future.

$$S.\texttt{CostImp}(l) = \max\left\{ S.c - S.c^{1\mathrm{st}}, S.c^{1\mathrm{st}} - S.c^{2\mathrm{nd}}, 2\frac{S.l^{1\mathrm{st}} - l}{S.s} \right\} \tag{2}$$

Our proposed priority metric is essentially the negative of the projected loss of $S$:

$$\mathcal{F}.\texttt{Priority}(S) = -(S.l^{1\mathrm{st}} - S.s \times b) \tag{3}$$

in which $b = \min(\max_{S \in \mathcal{F}.\mathbb{S}} S.\texttt{CostImp}(\mathcal{F}.l^*), B - \mathcal{F}.c)$. $\max_{S \in \mathcal{F}.\mathbb{S}} S.\texttt{CostImp}(\mathcal{F}.l^*)$ can be considered as the resource needed for every $S$ to have a better performance than the currently best performance $\mathcal{F}.l^*$. Our priority metric estimates the loss of each search thread if such an amount of resource (restricted by the budget left $B - \mathcal{F}.c$) is given. By considering both the search threads' current performance and potential improvement, it provides a fair trade-off between exploiting the currently-best and exploring the potentially-better choices.

---

**Algorithm 2** `SearchEvaluate1Step`

---

    **Inputs:** HPO problem $P$, search thread $S$, and $\mathcal{F}$
1: **if** $S$ is None **then** Construct $\mathbf{x}$ as follows: generate the controlled dimensions of $\mathbf{x}$ by adding Guassian noises on the corresponding dimensions of $\mathcal{F}.\mathbf{x}_0$ and for the rest of the dimensions sample uniformly at random from the search space $P.\mathcal{X}$.
2: **else** $\mathbf{x} \leftarrow S.\texttt{ProposeConfig}()$
3: **if** $S = S_0$ (i.e., $S$ is the global search thread) & $\mathbf{x} \notin \mathcal{F}.\mathcal{R}$ **then** $\mathbf{x} \leftarrow \texttt{invalid}$
4: **else** $l, c \leftarrow P.\texttt{LossFunc}(\mathbf{x}), P.\texttt{CostFunc}(\mathbf{x})$            ▷ Evaluate configuration $\mathbf{x}$
5: **if** $\mathbf{x} \neq \texttt{invalid}$ **then**
6:     $\texttt{BookKeeping}(S, \mathbf{x}, l, c, \mathcal{F})$ and update speed $S.s$,
7:     $\forall i \in P.D, S.x_i^{\min} \leftarrow \min\{\mathbf{x}_i, S.x_i^{\min}\}, \mathcal{F}.x_i^{\min} \leftarrow \min\{S.x_i^{\min} - \mathcal{L}.\Delta, \mathcal{F}.x_i^{\min}\}$,
8:     $\forall i \in P.D, S.x_i^{\max} \leftarrow \max\{\mathbf{x}_i, S.x_i^{\max}\}, \mathcal{F}.x_i^{\max} \leftarrow \max\{S.x_i^{\max} + \mathcal{L}.\Delta, \mathcal{F}.x_i^{\max}\}$
9: **Outputs:** $\mathbf{x}, l, c$

---

**Step 2: Config validator and evaluator (line 4-5 of Alg 1).** After a search thread (and a backup search thread) is selected, the next step is to propose the next configuration to try with the chosen search thread(s). Intuitively speaking, we consider generating the next configuration to try primarily according to the selected search thread $\tilde{S}$ whose priority is ranked the highest. But we set a guard rail for the global search thread as it may propose an unnecessarily high-cost configuration. We thus introduce a *config validator* to validate the configurations proposed by global search according to whether they are within the current admissible region of our framework $\mathcal{F}.\mathcal{R}$ (line 3 of Alg 2). A configuration marked as 'invalid' means that it is considered to be prone to incur unnecessarily high cost and will not be evaluated at this round. In this case, the selected backup search thread will be used to perform another round of `SearchEvaluate1Step` (line 5 of Alg 1) if it is a valid search thread (i.e., not none). In the case where the backup thread is none, we generate the new configuration according to line 1 of Alg 2. The config validator helps avoid potentially high-cost evaluation and thus avoid creating local search threads from high-cost points until necessary. It does not stick to local searches forever because the admissible region $\mathcal{F}.\mathcal{R}$ gets expanded. Note that according to the definition of $\mathcal{F}.\mathcal{R}$, only the controlled dimensions of the hyperparameter configurations are subject to validation check. If needed, a multi-fidelity pruning strategy can be used in this config evaluator component. Multi-fidelity pruning does not necessarily yield better performance. So the adoption of multi-fidelity pruning in `BlendSearch` is optional.

**Step 3: Search thread creator, updater and cleaner (line 6-11 of Alg 1).** If the newly proposed configuration is proposed by global search and it is not marked as 'invalid', we consider creating

a new local search thread using the proposed configuration as a starting point. To make sure the newly created local search thread is relatively good, we first check whether the proposed configuration's performance is better than at least half of the existing threads' performance (specified in the `CreateNewLSCondition`). If so, a new local search thread will be initialized and added to the active search thread pool $\mathbb{S}$. In `DeleteAndMergeLS`, we check whether a local search thread has converged according to the convergence condition of the specific local search method. If it is, the search thread will be removed from $\mathbb{S}$. In addition, we also go through all the local search threads to see whether the incumbent of a LS thread is reachable in one step by another LS thread with lower loss (ref. Appendix A). If so, the former LS thread will be deleted. After a configuration proposed by global search is evaluated, the observation tuple $(\mathbf{x}, l, c)$ is then used to update the model of the global search method through function `UpdateGSModel`. For example, when the global search method is a Bayesian optimization method, the model is the surrogate model used.

Due to page limit, detailed pseudocode for several of the straightforward functions mentioned in our framework are provided in Appendix A, including `CreateNewLSCondition`, `DeleteAndMergeLS`, `InitializeSearchThread` and `BookKeeping`.

## 4 EXPERIMENTS

We evaluate `BlendSearch` in tuning tabular machine learning libraries with an AutoML benchmark (Gijsbers et al., 2019), and in fine-tuning NLP models for text data. The AutoML benchmark consists of 39 tabular datasets that represent real-world data science classification problems. It includes datasets of all sizes, of different problem domains and with various levels of difficulty. As each dataset has 10 cross-validation folds, all the results reported in this paper are averaged over the 10 folds. With this benchmark, we are able to evaluate multiple HPO methods on a large number of datasets within a manageable computational budget, for tuning three machine learning libraries: XGBoost (Chen & Guestrin, 2016), LightGBM (Ke et al., 2017) and DeepTables[2]. The first two are popular libraries based on gradient boosted trees, and the third is an easy-to-use deep learning toolkit which utilizes latest research findings for tabular data. We chose them because gradient boosted trees and deep neural networks are the most frequent winners in data science competitions. We run experiments for XGBoost and LightGBM on the AutoML benchmark and report the results in Section 4.1. As a real application, we report an NLP model fine-tuning task for a production use case in Section 4.2. Finally, in Section 4.3 we perform ablation study to investigate the effectiveness of several important components of our framework. Due to page limit, we include the results for tuning DeepTables in Appendix B. We include the following baselines in our experiments.

- BO (Akiba et al., 2019) – the Bayesian optimization baseline. We choose a modern HPO library Optuna and use the TPE sampler because of its flexibility in handling mixed continuous and discrete space and good peformance reported in existing work (Falkner et al., 2018).
- LS (Wu et al., 2021) – the recent baseline of using local search with random restart, based on `FLOW`[2]. It is proved to be able to control cost effectively and outperform BO methods for numerical cost-related hyperparameter search.
- BOwLS (Gao et al., 2020) – the baseline of an existing approach of combining local search with BO, i.e., using BO to propose start points for local search.
- ASHA (Li et al., 2020), i.e., asynchronous successive halving – a state-of-the-art HPO method that uses multi-fidelity optimization and supports parallel tuning.

**Initialization setting.** For the local search method used in our framework, a low-cost initialization is needed to realize its unique advantages in controlling the cost. It is implemented via setting a low-cost initial value for each of the controlled dimensions. For example, among the hyperparameters tuned in LightGBM (shown in Table 2 of Appendix B), three hyperparameters, including 'tree num', 'leaf num' and 'min child weight' have initial values corresponding to the min or max values in their range, depending on whether they have a positive or negative correlation with the evaluation cost. It does not require the loss of the initial configuration to be low. Only one single low-cost initial value for each controlled dimension needs to be specified as input. To ensure a fair evaluation, we use the same low-cost initial point (if controlled dimensions exist) as the starting point of all the baselines.

---

[2]`https://github.com/DataCanvasIO/DeepTables`

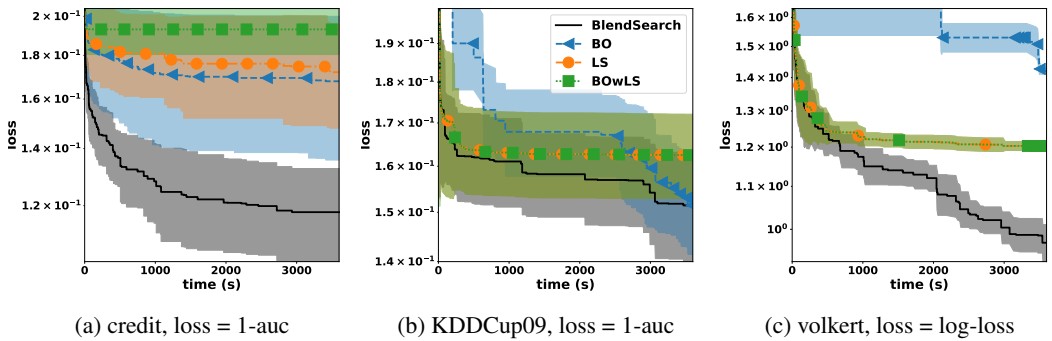

(a) credit, loss = 1-auc      (b) KDDCup09, loss = 1-auc      (c) volkert, loss = log-loss

Figure 3: Optimization performance curve for XGBoost. Lines correspond to the mean loss over 10 folds, and shades correspond to 95% confidence intervals. 1-auc is used for binary classification and log-loss is used for multi-class classification.

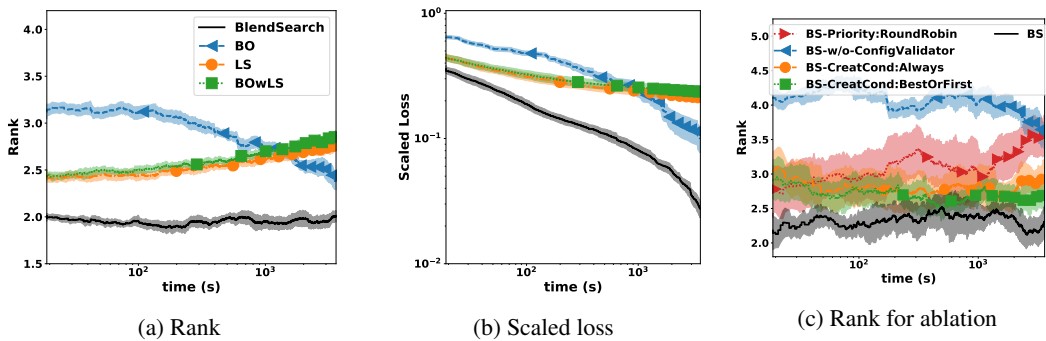

(a) Rank      (b) Scaled loss      (c) Rank for ablation

Figure 4: Aggregated rank and scaled loss on LightGBM and XGBoost. The scaled loss is obtained by min-max scaling using the max and min loss over all time across all the methods. Sub-figure (a) & (b) share the same legend, and BS is short for BlendSearch in sub-figure (c). Both rank and scaled regret are aggregated first across datasets then across 10 folds of data in each dataset.

## 4.1 TUNING XGBOOST AND LIGHTGBM

We tune a set of 9-dimensional hyperparameters (all numerical) in LightGBM and 11-dimensional hyperparameters (9 numerical and 2 categorical) in XGBoost. A detailed description of the search space can be found in Appendix B. In this section, we omit the multi-fidelity baseline method, i.e., ASHA, because we do not find a good 'fidelity' dimension that works well in tuning LightGBM and XGBoost (ref. additional results in Appendix B). We perform the evaluation on 37 out of the 39 datasets from the AutoML benchmark ('Robert' and 'Dionis' are excluded due to out-of-memory error and extremely long training time). The input budget $B$ (in terms of CPU time) is set to be 4 hours for the 3 largest datasets among the 37 datasets, and 1 hour for the rest. From the performance curves shown in Figure 3, we observe that BO tends to perform well on small datasets, e.g., 'credit' in Figure 3(a), where the 1h budget is sufficient. Under the same budget, it may perform badly on large datasets, e.g., 'volkert' in Figure 3(c), as it may try configurations which consume a very large portion of the budget at the early stage of the search. On the medium size dataset, e.g., 'KDDCup09' in Figure 3(b), the local search method is more efficient than BO in the early stage but is outperformed in the later stage. `BlendSearch` performs similarly to the better one between LS and BO in the early stage, and surpasses both of them in the later stage. We also observe that BOwLS performs similarly with LS (sometimes worse). This is because BOwLS needs to wait until a local search converges before proposing a new one. The aggregated result over all the test cases in Figure 4(a) & (b) is consistent with these observations.

**The interplay between local and global search in `BlendSearch`.** We investigated the dynamics of local and global search thread selection in `BlendSearch`. As a case study, we show the results in tuning XGBoost on two datasets in Figure 1(b) and Figure 5(a). The result in Figure 1(b) shows

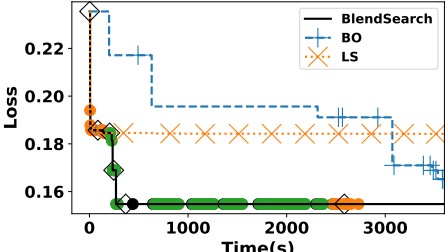
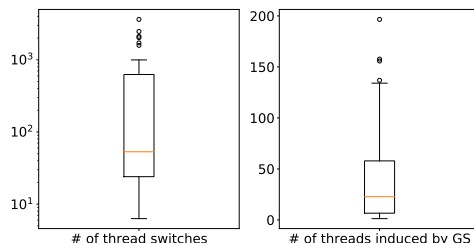

Figure 5: Search thread selection in `BlendSearch` in tuning XGBoost on dataset *KDDCup09*. In this figure (and Figure 1(b), which shows the result for dataset *Volkert*), each marker corresponds to one time of search thread selection in `BlendSearch`. The diamonds correspond to the global search thread, and the solid circles correspond to the local search threads (different colors for different threads).

Figure 6: The distributions of the # of thread switches and the # of local search threads induced by the GS (short for global search) in `BlendSearch` for tuning XGBoost. The result is aggregated over all the datasets evaluated in tunning XGBoot. The outliers with very large # of thread switches and threads induced by GS are from very small datasets, where the number of evaluations is large.

that the global search indeed plays a role after 1000s which contributes to `BlendSearch`'s better performance comparing to LS. The result in Figure 5(a) shows that although the first point suggested by the global search (at around 200s) does not yield significantly better performance immediately, the induced new local search thread (the thread in green circle) lead to a significantly better performance soon. These two figures together indicate that global search is not only responsible for directly finding better configurations, but also for creating local search threads that can achieve yet better performance. We show two statistics about the overall interplay between global and local search in `BlendSearch` on all the datasets evaluated for tuning XGBoost in Figure 6.

We provide additional experiments for tuning XGBoost and LightGBM in Appendix B, including a comparison with ASHA under different settings of fidelity, and an empirical study about the effect of low-cost initialization.

**Takeaway.** (1) `BlendSearch` is able to overcome the limitations of BO and LS and at the same time, inherit their advantages. By blending BO and LS, `BlendSearch` is able to outperform both on this large collection of datasets over time. (2) The interplay between global and local search indeed contribute to `BlendSearch`'s good performance.

## 4.2 TRANSFORMER-BASED NLP MODEL FINE-TUNING

This section presents an application of economical hyperparameter optimization to an NLP model fine tuning task used in a large software company. It starts from a large transformer-based model Turing-NLRv2 with 24 transformer layers, each with 16 attention heads and hidden-dimension 1024 totaling 340M parameters. It is pretrained on English Wikipedia (2,500M words) and BookCorpus (800M words), and uses byte-pair encoding[3]. This pre-trained model is then fine-tuned (Dai & Le, 2015) for use in multiple production scenarios, including sequence classification, named entity recognition and question answering. The fine-tuning procedure is performed for a dozen separate tasks, and is repeated on a regular cadence, typically every few weeks. We focus our experiment on a single sequence classification fine-tuning task where the objective is to label a document (consisting of one or more sentences) with one of five possible classes. For fine-tuning this model we introduce a classification layer with $1024 \times 5 = 5120$ additional weight parameters, randomly initialized. The dataset used for training consists of 52K labeled examples, which we split 80/20 for training/validation. The objective to maximize is the f1-score obtained on the validation set of 10.4K labeled documents. Selecting hyperparameters for fine-tuning this model has been a manual process that typically takes a data scientist a few days.

---

[3]`msturing.org`

Table 1: Results of language model fine-tuning.

| method | wall-clock | GPU-hours | f1 |
|---:|---|---|---|
| Manual | (10-20 working hours, spanned to a few days) | 40-80h | 0.645 |
| BlendSearch | **3h** | **12h** | **0.652** |
| ASHA-1 | 6h | 24h | 0.624 |
| ASHA-16 | 6h | 384h | 0.640 |

In our experiment we use a 6-dimensional search space (4 numerical and 2 categorical). A detailed description of all the hyperparameters tuned, including their ranges, can be found in Table 5 in Appendix B. We compare to ASHA and let it use 16 VMs with 4 NVIDIA Tesla V100 GPUs on each VM. We run ASHA with 16 concurrent jobs for 6 hours in wallclock time. That amounts to $4 \times 16 \times 6 = 384$ GPU hours of hardware cost. We run `BlendSearch` on a *single* VM of the same configuration for 3 hours, which uses 12 GPU hours in total. For comparison, we include ASHA-1 using the same single VM. The results are summarized in Table 1.

**Takeaway.** Not only `BlendSearch` finds a more accurate model than both ASHA and manual effort, it does so faster and consumes only 3% of the resources of ASHA-16.

### 4.3 ABLATION STUDY

To investigate the effectiveness of several important modules of our framework (colored in orange in Figure 2), we perform an ablation study in tuning XGBoost on a random subset of the datasets (one third of the datasets mentioned in Section 4.1). We show the aggregated rank of different variants of our method in Figure 4(c). Specifically, we study the following three modules of the framework. **(1) Priority metric.** 'BS-Priority:RoundRobin' uses the round-robin policy in the search thread selector. **(2) Config validator.** 'BS-w/o-ConfigValidator' skips the validity check of the proposed configuration. **(3) Create new thread condition.** In 'BS-CreateCond:Always' and 'BS-CreatCond:BestOrFirst', the following two conditions are used as the condition for creating new local search threads respectively: always create, and create a new thread only when the loss is better than all the existing search threads' loss or there is no local search thread yet.

From this ablation study, we have the following observations: (1) Making round-robin selection is worse than doing selection using our designed priority metric. Round-robin's relative performance becomes worse in the later stage of the search because it cannot avoid bad-performing search threads. (2) The config validator is also vital in our framework. (3) Overall, the conditions for creating new threads has a smaller impact on our method comparing to the other designs studied. 'Better than half' condition used by default tends to perform the best.

## 5 EXTENSION AND FUTURE WORK

In the low-resource scenario which is targeted by this paper, each single trial is not resource-saturated if we spend all resources in it. So we do not recommend parallel trials in this low-resource scenario. In the case where more resources than the maximum resource each single trial can consume are available, our framework can be extended by running the trials from different search threads on multiple workers. For example, if there are additional workers available, we can keep invoking the search thread selector (but skip the local search threads that have $O(d)$ trials running). Our design of having multiple independent local search threads naturally allows efficient asynchronous parallel trials. The design of utilizing existing global optimization methods allows existing easy-to-parallelize global optimization (such as random search or batch versions of BO) to be plugged in. The prioritization of search threads is still useful as long as the maximal concurrent number of trials divided by the number of search threads is smaller than $O(d)$. Since our method can be used together with multi-fidelity pruning methods, it can naturally inherit the asynchronous resource scheduling when used in the parallel setting. Parallelization is now supported in the latest version of `BlendSearch`'s implementation.

In this work, we show the effectiveness of `BlendSearch` through extensive empirical study. As future work, it is worth studying the theoretical properties of `BlendSearch`, including theoretical guarantees about its convergence rate and total resource consumption.

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

## A    More details about BlendSearch

We provide detailed pseudocode of the sub-algorithms used in our framework.

---

**Algorithm 3** `SelectThread`

---

    **Inputs:** Framework-level state $\mathcal{F}$ which keeps a list of candidate search thread $\mathcal{F}.\mathbb{S}$ and a priority function $\mathcal{F}.\texttt{Priority}$.

1:   $\tilde{S} = \arg\max_{S \in \mathcal{F}.\mathbb{S}} \mathcal{F}.\texttt{Priority}(S)$

2:   $\tilde{S}_{\text{bak}} = \begin{cases} \arg\max_{S \in (\mathcal{F}.\mathbb{S} \setminus S_0)} \mathcal{F}.\texttt{Priority}(S) & \mathcal{F}.\mathbb{S} \setminus S_0 \neq \emptyset \\ S_0 & \mathcal{F}.\mathbb{S} \setminus S_0 = \emptyset, \end{cases}$

    **Outputs:** $\tilde{S}, \tilde{S}_{\text{bak}}$

---

**Algorithm 4** `BookKeeping`

---

    **Inputs:** Search thread $S$, one entry of observation $\mathbf{x}, l, c$ generated by $S$, and $\mathcal{F}$.

1:   $S.c \leftarrow S.c + c$

2:   **if** $S$ is a local search thread **then** Update $\{S.x_i^{\min}\}_{i \in [d]}$, and $\{S.x_i^{\max}\}_{i \in [d]}$

3:   **if** $l < S.l^{\text{1st}}$ **then**

4:      $S.l^{\text{2nd}}, S.c^{\text{2nd}} \leftarrow S.l^{\text{1st}}, S.c^{\text{1st}}; S.l^{\text{1st}}, S.c^{\text{1st}} \leftarrow l, S.c$; and $S.\mathbf{x}^{best} \leftarrow \mathbf{x}$

5:      **if** $l < \mathcal{F}.l^*$ **then** $\mathcal{F}.l^* \leftarrow l$

---

**Algorithm 5** `InitializeSearchThread`

---

    **Inputs:** Search method $\mathcal{L}$ or $\mathcal{G}$, problem $P$ and initial point $(\mathbf{x}, l, c)$.

1:   $S.\pi = \mathcal{L}.\pi$ or $\mathcal{G}.\pi$

2:   Initialize bookkeeping information of $S$: $S.l_1 = S.l_2 = l$, $S.c^{\text{1st}} = S.c^{\text{2nd}} = S.c = c$, and $S.x_i^{\min} = S.x_i^{\max} = \mathbf{x}_i$ for $i \in P.D$

---

**Choice of $\mathcal{L}.\Delta$ used in constructing the 'admissible' region $\mathcal{F}.\mathcal{R}$.** We normalize each numeric hyperparameter into [0,1]. The setting of $\mathcal{L}.\Delta$ is decided by the local search method. For the local search method we chose, $\mathcal{L}.\Delta$ is a constant corresponding to the initial stepsize (fixed as 0.1).

**Definition of `ReachableInOneStep` in Alg 7.** We define local search thread $S_1$ to be reachable by $S_2$ if the distance between their incumbents is no larger than the maximal distance between the next proposal of $S_2$ and the incumbent of $S_2$. In the local search method we used, the incumbent is the currently best config, and the maximal distance is equal to the stepsize.

## B    More details about experiments and additional results

### B.1    Experiment setup

**Settings of BO and LS.** For BO, we use implementation from Optuna 2.0.0 (`https://optuna.readthedocs.io/en/stable/index.html`) with default settings for TPE sampler. For LS, we follow the implementation guidelines from Wu et al. (2021). After a local search thread is created from a particular starting point, we fix the categorical dimensions and only search for numerical dimensions in that local search thread. A local search thread $S$ is considered to have converged (corresponding to $S.\texttt{converged}()$ in Algorithm 7) once the stepsize of the local search thread is smaller than a lower bound introduced by Wu et al. (2021).

**Experiments in tuning XGBoost and LightGBM.** The XGBoost and LightGBM experiments are performed in a server with Intel Xeon E5-2690 v4 2.6GHz, and 256GB RAM. A full list of hyperparameters tuned and their ranges can be found in Table 3 and Table 2. The search space for numerical hyperparameters aligns with the search space used in (Wu et al., 2021). On the same fold, the same random seed is used for LS, BO and BS. Experiments on different folds use different random seeds.

**Experiments in NLP model fine tuning.** For ASHA, we set min and max epochs as 1 and 16, and reduction factor 4.

---

**Algorithm 6** `CreateNewLSCondition`

---

    **Inputs:** $l, \mathcal{F}$.
    **Outputs:** $|\mathcal{F}.\mathbb{S}| = 1$ or $l \leq \text{Median}(\{S.l^{\text{1st}}\}_{S \in \mathcal{F}.\mathbb{S} \setminus S_0})$

---

---

**Algorithm 7** `DeleteAndMergeLS`

---

    **Inputs:** $S, \mathcal{F}$
1: **if** S.`converged()` **then**
2:     $\mathcal{F}.\mathbb{S} \leftarrow \mathcal{F}.\mathbb{S} \setminus S$,
3:     $\mathcal{F}.\mathcal{R}.x_i^{\min} \leftarrow \mathcal{F}.\mathcal{R}.x_i^{\min} - \mathcal{L}.\Delta$, and $\mathcal{F}.\mathcal{R}.x_i^{\max} \leftarrow \mathcal{F}.\mathcal{R}.x_i^{\max} + \mathcal{L}.\Delta, \forall i \in D'$
4: **else**
5:     **for** $\forall S' \in \mathcal{F}.\mathbb{S} \setminus S$ **do**
6:         **if** $S \in S'.$`ReachableInOneStep()` $\& \ S'.l < S.l$ **then**
7:             $\mathcal{F}.\mathbb{S} \leftarrow \mathcal{F}.\mathbb{S} \setminus S$
8:             break
9:         **else if** $S' \in S.$`ReachableInOneStep()` $\& \ S.l < S'.l$ **then** $\mathcal{F}.\mathbb{S} \leftarrow \mathcal{F}.\mathbb{S} \setminus S'$

---

**Result aggregation details.** Aggregated rank in Figure 4(a)&(c) and 12(c) is calculated as follows: (1) per dataset per fold, the method is ranked based on the loss on validation set at each second (x-axis), starting from when there is at least one finished config evaluation in any method; (2) the rank is then averaged across datasets per fold; (3) we finally compute the average rank (line) and confidence interval (shaded area) across 10 folds. Scaled loss in Figure 4(b) is calculated similarly. Per dataset per fold, min-max scaling is applied on each method using the maximum and minimum loss along the whole performance curve across all methods.

### B.2   Additional experimental results on LightGBM and XGBoost

**More performance curves on LightGBM and XGBoost.** The performance curves for tuning LightGBM on 3 representative datasets with an 1h budget are shown in Figure 7. We observe that the performance of LS is quite good (comparing to BO), especially on large datasets. This result is consistent with the results reported in (Wu et al., 2021), where all the hyperparameters for tuning are numerical. In our experiment of XGBoost tuning, we include categorical hyperparameters. LS performs worse in this case because the introduction of categorical hyperparameters amplifies the local search method's limitation of being trapped in local optima. The observations about `BlendSearch` for LightGBM are similar to XGBoost tuning. The performance curves on the three large datasets with a 4h budget are shown in Figure 8 and 9, where similar conclusions can be drawn.

**Multi-fidelity.** We compare BO and `BlendSearch` with the multi-fidelity baseline ASHA for tuning LightGBM and XGBoost in Figure 10. In this experiment, we tried two choices of fidelity dimensions with ASHA, including number of iterations and sample size (the sample size begins with 10K, so small datasets are excluded) respectively. The results show that the multi-fidelity baseline overall perform no better than BO and are significantly worse than `BlendSearch`.

Table 2: Hyperparameters tuned in LightGBM.

| hyperparameter | type | range | init |
|---|---|---|---|
| tree num | int | [4, min(32768, # instance)] | 4 |
| leaf num | int | [4, min(32768, # instance)] | 4 |
| min child weight | float | [0.001, 20] | 20 |
| learning rate | float | [0.01, 0.1] | random |
| subsample | float | [0.6, 1.0] | random |
| reg alpha | float | [1e-10, 1.0] | random |
| reg lambda | float | [1e-10, 1.0] | random |
| max bin | int | [7, 1023] | random |
| colsample by tree | float | [0.7, 1.0] | random |

Table 3: Hyperparameters tuned in XGBoost.

| hyperparameter | type | range | init |
|---|---|---|---|
| tree num | int | [4, min(32768, # instance)] | 4 |
| leaf num | int | [4, min(32768, # instance)] | 4 |
| min child weight | float | [0.001, 20] | 20 |
| learning rate | float | [0.01, 0.1] | random |
| subsample | float | [0.6, 1.0] | random |
| reg alpha | float | [1e-10, 1.0] | random |
| reg lambda | float | [1e-10, 1.0] | random |
| colsample by level | float | [0.6, 1.0] | random |
| colsample by tree | float | [0.7, 1.0] | random |
| booster | categorical | {gbtree, gblinear} | gblinear |
| tree method | categorical | {auto, approx, hist} | random |

Table 4: Hyperparameters tuned in DeepTables.

| hyperparameter | type | range | init |
|---|---|---|---|
| early stopping rounds | int | [1, max(min(1.5M/# instance),150),10)] | 10 |
| batch size | int | [16, 1024] | 512 |
| dropout | float | [0, 0.5] | 0.1 |
| learning rate | float | [1e-4, 3e-2] | 3e-4 |
| dense dropout | float | [0, 0.5] | 0.1 |
| net | categorical | {DCN, dnn_nets} | random |
| auto discrete | boolean | {False, True} | random |
| apply gbm features | boolean | {False, True} | random |
| fixed embedding dim | boolean | {False, True} | random |

Table 5: Hyperparameters tuned in fine-tuning Turing language model.

| hyperparameter | type | range | init |
|---|---|---|---|
| learning rate | float | [1e-6, 1e-3] | random |
| hidden dropout probability | float | [0.05, 0.4] | random |
| warmup proportion | float | [0.2, 0.4] | random |
| batch size | categorical | {16, 32} | 32 |
| epochs | int | [1, 16] | 1 |
| learning rate scheduler | categorical | {Warmup linear decay polynomial, Warmup linear, Warmup linear decay exponential} | random |

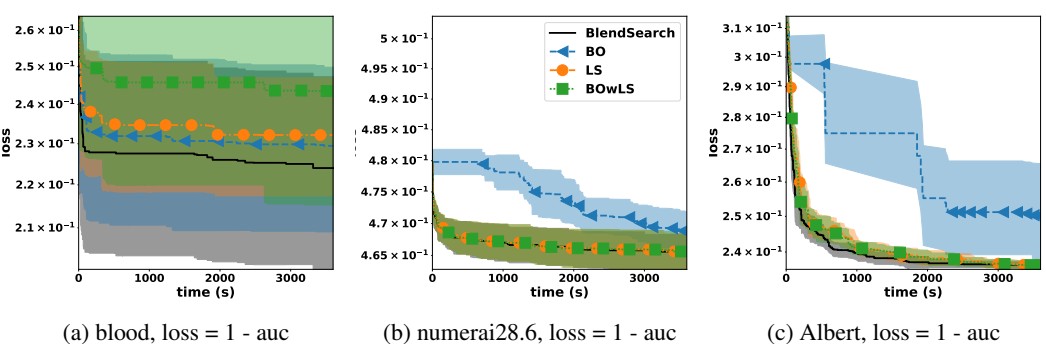

(a) blood, loss = 1 - auc     (b) numerai28.6, loss = 1 - auc     (c) Albert, loss = 1 - auc

Figure 7: Optimization performance curve for LightGBM (1h).

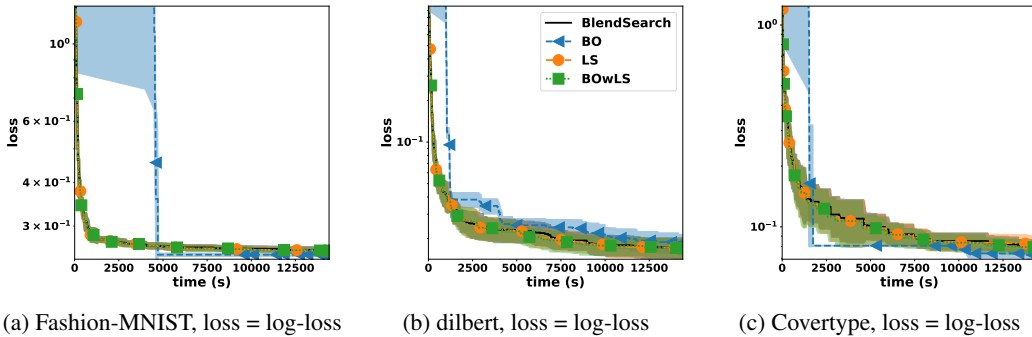

(a) Fashion-MNIST, loss = log-loss    (b) dilbert, loss = log-loss    (c) Covertype, loss = log-loss

Figure 8: Optimization performance curve for LightGBM (4h).

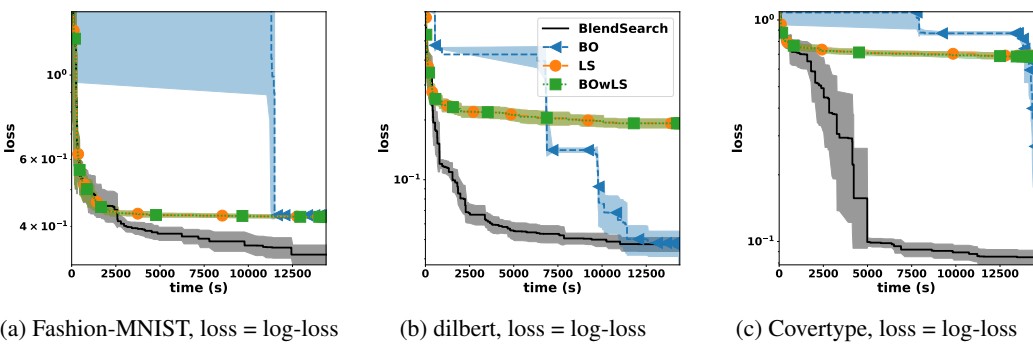

(a) Fashion-MNIST, loss = log-loss    (b) dilbert, loss = log-loss    (c) Covertype, loss = log-loss

Figure 9: Optimization performance curve for XGBoost (4h).

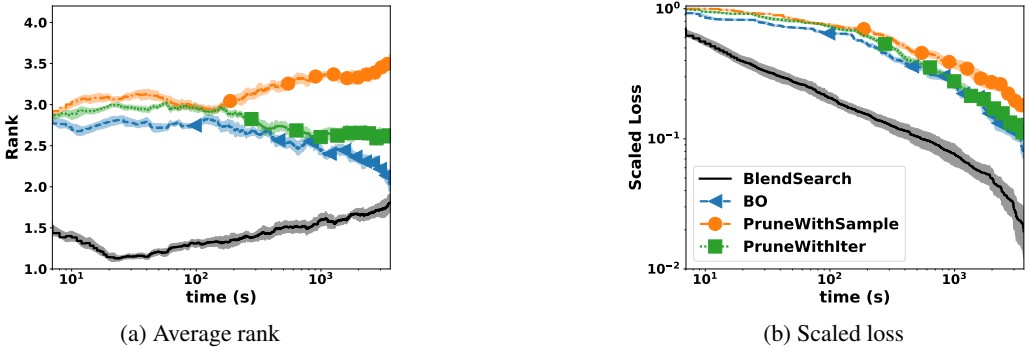

(a) Average rank    (b) Scaled loss

Figure 10: Aggregated results on LightGBM and XGBoost. 'PurneWithSample' and 'PruneWith-Iter' represent ASHA using sample size and iteration number as resource dimension respectively. BO and `BlendSearch` are the same as those in Figure 4.

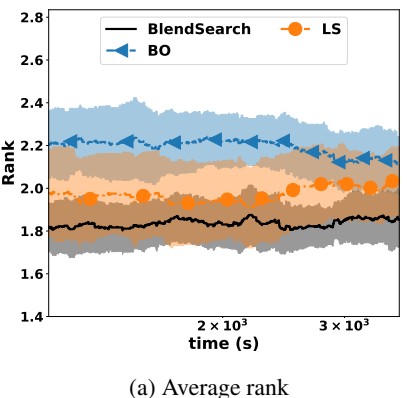

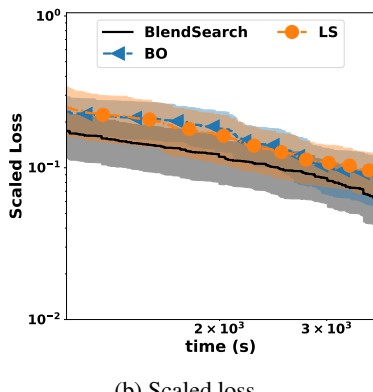

(a) Average rank

(b) Scaled loss

Figure 11: Aggregated rank and scaled loss on LightGBM with random initialization.

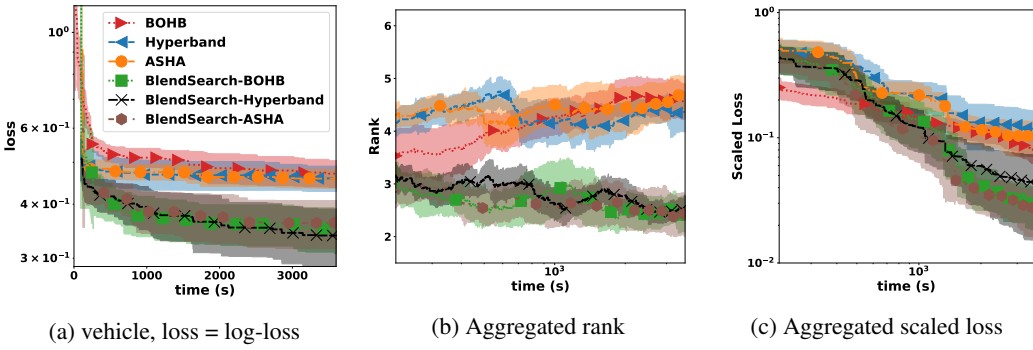

(a) vehicle, loss = log-loss

(b) Aggregated rank

(c) Aggregated scaled loss

Figure 12: Performance curves on *cane*, aggregated rank and aggregated loss on DeepTables.

**Ablation study on the low-cost initialization.** In this work, we use low-cost initialization for the controlled dimensions of the hyperparameters. Although such information is fairly easy to obtain, we investigate our method's robustness when no controlled dimension is provided. We test `BlendSearch` in a controlled dimension agnostic setting: there are still hyperparameters with heterogeneous cost, but the controlled dimensions and a low-cost initial point is not specified as input. In such scenarios, `BlendSearch` will use random initialization and the config validator always returns 'yes'. We compare `BlendSearch` with local search and BO under such a setting using the same random initial point. In Figure 11 we report the results including the aggregated rank and scaled loss on LightGBM across half of the datasets mentioned in Section 4.1 in Figure 11(a) & (b). The results show that even if `BlendSearch` is agnostic to the controlled dimensions and a random initialization is used, it is still able to outperform both the local search method and BO.

### B.3 Tuning DeepTables.

In this experiment, we tune 9-dimensional hyperparameters (5 numerical and 4 categorical as detailed in Table 4) in DeepTables. Since the training of deep neural networks are more time-consuming than that of XGBoost, we run experiments for DeepTables on the datasets where they are worse than the best known performance in the benchmark, including *'shuttle','cnae','mfeat','vehicle','phoneme', 'kc1'*. All experiments for DeepTables are performed in a server with the same CPU, 110GB RAM, and one Tesla P100 GPU. A full list of hyperparameters tuned and their ranges can be found in Table 4.

Recall that we mentioned multi-fidelity pruning strategies could be incorporated into `BlendSearch` in the config evaluator component. In this experiment, we are particularly interested showing the performance of `BlendSearch` when combined with multi-fidelity methods. To this end, we include the three state-of-the-art multi-fidelity methods, including BOHB (Falkner

et al., 2018), ASHA (Li et al., 2020), and asynchronous HyperBand (Li et al., 2017; 2020) which are shown efficient for tuning deep neural networks and the `BlendSearch` based on each of them. We use the following libraries for baselines: For BOHB, we use HpBandSter 0.7.4 (`https://github.com/automl/HpBandSter`). For ASHA and asynchronous HyperBand, we use implementations from Optuna 2.0.0. In all the methods compared, including both existing methods and variants of `BlendSearch`, the number of training epochs is used as the fidelity dimension, with maximum epochs set to be 1024, reduction factor set to be 3, and minimum epochs 4. For ASHA, we set the minimum early stopping rate to be 4 (we adopted this setting as it yields better performance comparing to the default setting, i.e., 0). The number of training epochs is used as the fidelity dimension.

`BlendSearch` incorporates existing multi-fidelity methods in the following way: Each config, either proposed by global search or local search, uses the same schedule to increase the fidelity and check its pruning condition. For example, when ASHA (Li et al., 2020), i.e., asynchronous successive halving, with a reduction factor of $\eta$, is used as the pruning strategy, after each config is evaluated by a certain fidelity, it is compared with other configs already evaluated by the same fidelity. The config will be pruned if its loss is ranked in the worst $1/\eta$. Otherwise, the fidelity is multiplied by $\eta$. In addition to the original pruning conditions specified by the multi-fidelity method, a configuration will also be pruned at a particular fidelity level where no pruning is performed yet, and the configuration does not yield superior performance (comparing to the currently-best performance) when evaluated at that fidelity level.

We present the performance of all compared methods for tuning DeepTables in Figure 12. Figure 12(a) shows the learning curves on dataset *cane* with budget 1h. Figure 12(b) and (c) show the aggregated rank and loss on all the 6 datasets within budget 1h. The performance of multi-fidelity methods are significantly improved when used in our `BlendSearch` framework.

