# OpenReview forum: "ECONOMIC HYPERPARAMETER OPTIMIZATION WITH BLENDED SEARCH STRATEGY"
_ICLR.cc/2021/Conference — ICLR 2021 Poster_

### Official Review · AnonReviewer4 · 2020-10-26
**A complex algorithm with some impressive results, but lacking rigorous and/or intuitive justification.**

**Rating:** 7
**Confidence:** 4

**Review:**

This paper proposes a new algorithm for black-box optimization (BlendSearch) that combines global search methods together with local search methods. The stated goal is to ensure convergence to the global optimum, while avoiding configurations with a high cost (e.g. that take a long time to evaluate).

Pros:
- Section 2 (related work) is well-written and quite comprehensive.
- Extensive experiments on different applications (XGBoost, LightGBM, DeepTables, fine-tuning NLP) which show promising results.

Cons:
- The paper is lacking intuitive explanations for how and why the algorithm works so well.
- Theoretical and/or experimental justifications are needed for some of the key statements in the paper.

My main criticism of the paper is regarding Section 3 and how the BlendSearch algorithm is described and presented. Clearly, the algorithm itself is a complex piece of engineering, and while that by itself is not a problem, very little intuition is provided to help the reader understand why the different design choices were made. Furthermore, it is not very clear to me how or why the fundamental claims made about the algorithm are true. For instance, it is stated that the algorithm will converge to the global optimum given sufficient budget. Is it guaranteed under all conditions? Maybe this needs a theoretical proof or maybe it is somehow obvious by construction, but either way the paper should explain this clearly. Additionally, it is not very clear why the config validator step helps to avoid configurations with a high cost: the process is described in technical detail but what is the intuition here? Fundamentally: why does local search help to avoid evaluating high cost configurations?

In Section 1 it is stated that multi-fidelty methods (e.g. successive halving or Hyperband) can only be used when cheap proxies for the “accuracy assessment” exist. However,  isn’t it always possible to construct such a proxy by taking a random subsample of the training data (e.g. as suggested in the Hyperband paper). In Section 4.1 the authors say that multi-fidelity methods were not considered as a baseline when tuning XGBoost and LightGBM, because they tried using both subsample size as well as number of iterations and in both cases it “hurt the performance”. It would improve the paper to include some experimental results to justify this statement. While of course it is true that a hyper-parameter configuration that works well on a small subsample (or small number of boosting iterations) may not work well on the full dataset (or with a large number of iterations), similar arguments can be made about the number of epochs when training neural networks (e.g. optimal learning rate for small number of epochs is not necessarily the optimal learning rate for a large number of epochs). This does not stop multi-fidelity methods being widely used to tune neural networks.

Another area where I think the paper could be improved is with how the authors deal with parallelism. While there does seem to be a paragraph in the Appendix stating that BlendSearch can be parallelized, it is not discussed in the main manuscript. Methods like Hyperband and successive halving and in the extreme case, random search, admit very high degree of parallelism. Because of this, in many real applications they are often preferred over more complex methods. While the results of Section 4.3 indicate that BlendSearch using 1 worker can out-perform asynchronous successive halving (ASHA) using 16 workers which is a nice result, it is not clear whether any parallelism was employed for the results in Section 4.2. If no parallelism was used, does ASHA even make sense (e.g. without any parallelism there is no need for the ‘asynchronous’ aspect)?. The paper could be made stronger by showing how the performance of BlendSearch improves as more workers are added, and how it compares to schemes like Hyperband and AISA with the same number of workers.

Additional comments:
- Figure 1 does not seem to be referenced from anywhere the text. There are also no error bars shown on the plots. Given that the algorithms being compared are highly sensitive to their initial conditions, it is not possible to make statements without running multiple times with different random seeds.
- In Section 4.2, the author state that ‘BlendSearch-HyperBand’ uses random search for global search and HyperBand pruning strategy. However, the concept of a pruning strategy is not discussed anywhere in the manuscript. It is hard to understand what this means.
- The notation used in Section 3 to group certain variables as “attributes” (e.g. P.CostFunc(.)) is quite non-standard and a bit hard to read, but maybe this is a personal preference.
- In Section 3 could be improved by carefully defining what exactly what is meant by a ‘thread’ in this context.
- In the experimental section, were the optimization algorithms re-run using different random seeds, or are all statistics generated over multiple folds with the same optimization seed? If the latter, the results could be severely biased if one algorithm uses a ‘lucky’ seed that results in a good sequence of configurations that work well across most datasets.

---

> ### Author Response · Authors · 2020-11-20
> **Response to AnonReviewer4**
>
> We thank the reviewer for the constructive comments. Please find our response to all the comments below.
>
> 1. The intuition of our method, and the roles of local search and config validator in helping achieve low-cost.
> Our method can achieve low cost because of the following designs: (1) The first local search thread begins with a cheap config and restricts its proposals to be close to the incumbent. Assuming cost continuity in the search space, configs proposed by the local search method have bounded total cost as proved by Wu et al. (2020); (2) With the same cost continuity assumption, the config validator helps avoid potentially high-cost evaluation and thus avoid creating local search threads from high cost points until necessary; and (3) Our search thread selection strategy prioritizes different search threads according to their efficiency, the intuition of which is added in Section 5.
> 2. Convergence to global optima.
> There are indeed two conditions for BS to converge to global optima: (1) the global search can converge to global optima with sufficient budget; (2) each local search thread converges (not necessarily to a local or global optimum) in finite time. The first condition is quite basic for global search methods, and the second is quite basic for local search methods.
> 3. Multi-fidelity methods for trees.
> We understand your concern. We have added experiments in Section 5 to support our observations.
> 4. Parallelism.
> No parallel trial is used in Sec 4.2. The asynchronous aspect of ASHA still makes sense in that setting as it allows high fidelity to be used before many configs of low fidelity have to be evaluated. Under this sequential setting, in addition to ASHA, we also compared with other resource scheduling methods including Hyperband and BOHB.
> We understand your interest in high-resource scenario. In this work, we are primarily targeting the low-resource setting with the general goal of making HPO more economical. In the low-resource setting, it is more desirable to use the limited resource in one trial at a time instead of splitting it into multiple trials (Erickson et al., 2020). So we have not prioritized the parallel version of BlendSearch for the sake of step-by-step progress. Implementing and testing an asynchronous parallel version of BlendSearch is indeed a next step which requires a good amount of engineering work. We updated the discussion about how to parallelize BlendSearch in the appendix. The design of having multiple mostly independent local search threads naturally fits the parallel and asynchronous scenario. The design of utilizing existing global optimization methods allows existing easy-to-parallelize global optimization such as random search or TurBO (Eriksson et al., 2019) to be plugged in. Since our method can be used together with ASHA, it can naturally inherit the easy-to-parallelize and asynchronous resource scheduling in ASHA when used in the parallel setting.
> 5. Response to detailed comments.
>      - Figure 1. Figure 1 is plotted based on results from a single run because we want to visualize the concrete local search and global search steps taken in BlendSearch. It is a demonstrative figure mainly used for providing intuitions. All the other results reported on the AutoML benchmark except this demonstrative figure are from multiple (10) runs.
>      - Pruning strategy. A pruning strategy refers to a strategy that can allocate different levels of resources (according to a resource schedule) to do the configuration evaluation and early-stop (i.e., prune) unpromising configs according to the intermediate results obtained. For example, ASHA is a pruning strategy, which executes in the following way: After each config is evaluated by a certain fidelity, this config is compared with other configs already evaluated by the same fidelity and pruned if its loss is in the worst 1/ReductionFactor.  If the config is not pruned, the fidelity of it will be multiplied by the ReductionFactor. In our results, the name `BlendSearch-X' means that BlendSearch uses the resource schedule produced by pruning strategy X. Each config, either proposed by global search or local search, uses the same schedule to increase the fidelity and check the pruning condition.
>      - Notations. Thanks for the suggestion.
>      - Search thread. A search thread is an instance of a GS or LS method, each with its own search trajectory.
>      - Random seed. Experiments on different folds use **different** random seeds. Further, for experiments on the same fold, the same random seed is used for different methods.
>
>
> Reference:
> Erickson, N., Mueller, J., Shirkov, A., Zhang, H., Larroy,P., Li, M., and Smola, A.   Autogluon-tabular:  Robust and accurate automl for structured data. arXiv:2003.06505, 2020.

---

### Official Review · AnonReviewer2 · 2020-10-26
**Next-step in blending local & global schemes for HPO; strong empirical performance; raises significant technical questions**

**Rating:** 6
**Confidence:** 5

**Review:**

The proposed BlendedSearch (BS) presents an intuitive next step in the combination of global and local search schemes for hyper-parameter optimization (HPO). Global search schemes are widely used for HPO but can suffer from large HPO times since their vanilla forms do not account for function evaluation costs. Local search schemes are usually not widely used for HPO but seem useful if the goal is to restrict the search to a region of the search space where the function evaluation costs do not grow drastically. The proposed BS interleaves global and local search steps to ensure that the global search does not go into regions of high evaluation costs while also avoiding being stuck in local minima.

The empirical evaluation of BS against various baselines demonstrate that BS has a better anytime performance relative to both local and global schemes. BS is also able to leverage multi-fidelity HPO schemes for reduced evaluation times and further improve upon them. The empirical evaluations also highlight the effect of the different parts of the proposed algorithm through an ablation study.


However, the proposed algorithm raises a lot of questions that are not addressed in this paper.

Most importantly, in my opinion, the empirical advantage of this proposed scheme is tied to how low-cost (and maybe low-loss) the initialization is from where the global and the local searches start. The initializations for the empirical evaluations are chosen to be low-cost (and maybe low-loss) for BS -- it is not completely clear from the main paper how these initializations were chosen and how (if at all) the considered baselines leveraged these initializations. It is not clear how BS would perform if initialized with a high-cost, high-loss point. It seems to me that having low-cost initialization(s) would also benefit something like GPEIPS -- instead of picking a single initial point, the GPEIPS could be initialized (similar to the proposed scheme) with a set of low-cost evaluations -- this would counter the issue in GPEIPS of requiring high-cost evaluations initially.

Another unknown is the interplay between the global and local search -- it is not clear whether the proposed scheme really leverages both global and local searches. In this current setup, where the local search threads focus on local "low-cost" regions, it seems plausible that the global search constantly get rejected repeatedly and never really used much (beyond initiating a few local threads). Given the "exploitation" evaluations obtained from the clearly favored local search threads, the current surrogate model (such as the GP regressor) would probably be in the "exploration" mode (trying out regions of the search space not part of the valid bounding box), which will often just get classified as invalid. For points to be valid, the $\Delta_i$ for the local searches need to be large, but the choice of these $\Delta_i$s are not discussed in this paper.

Moreover, the MBB of the union of the all the local thread regions does not imply that the evaluation costs stay low for anything in that region for the global thread to pick without additional assumptions on the cost structure. It might be good to justify such a choice. The ablation study shows that the anytime performance is heavily reliant on such a "ConfigValidator" that keeps the search in low cost regions, but this ties back to the question about how the search gets initiated and how that would affect the performance of BS. It is not technically clear how the current algorithm can recover if started from high-cost region.


Beyond this, there are various other questions which makes me think that we are not understanding the technical reason for the strong empirical performance, which makes me score this as a borderline paper.


Further questions:

- The $\Delta_i$ in the local search drives where the global search can look, so it is not clear how best to set that value (especially given the varying types and ranges for the hyper-parameters).
- Also it is not clear how local search is accomplished with mixed continuous-integer-categorical hyper-parameters.
- Why is "Trust region BO" (Eriksson et al., 2019) not applicable where the local models have "low-cost" initializations similar to the low-cost initializations used for BS?
- As per notation, given a budget $B$, the goal of the problem is to
 $$ \min_x P.LossFunction(x)  \mbox{ subject to } G(\pi) \leq B,$$
 not necessarily minimizing $G(\pi)$.
- In Fig 4(a), where evaluations are not costly, how is it that BS is outperforming the BO? What is the intuition behind that? Shouldn't BO be the best? Fig 1(b) makes more sense -- eventually BO and BS have similar performance but BS has improved anytime performance.
- Regarding "performance improvement speed S.s", it is unclear why speed is set to highest when the formula suggests it should be 0. This needs better clarification.
- Again, pertaining to the definition of performance improvement speed, if $S.l^{2nd} = S.l^{1st}$, isn't $S.c^{1st} = S.c^{2nd}$ by definition? What does that mean for the definition of speed?
- Formula in eq(2) could use more justification -- intuitively using the notion of "speed", we would have $S.s = (S.l^{1st} - l) / (S.CostImp(l) + S.c - S.c^{1st})$, implying $S.CostImp(l) = (S.c^{1st} + (S.l^{1st} - l) / S.s) - S.c$.
- It appears that the "Priority(S)" is some form of a UCB over a local thread -- it would be good to formally discuss how (if at all) this priority is connected to well established notions of UCB/EI.
- In "Step 3: Search thread creator....", what is precisely meant by "incumbent of a LS thread is reachable by another LS thread"
- The paper does not clearly discuss how the fidelity of the global search (handled by HyperBand, BOHB, etc) translates to the fidelity of the local searches when leveraging multi-fidelity schemes

---

> ### Author Response · Authors · 2020-11-20
> **Response to AnonReviewer2**
>
> We thank the reviewer for the insightful comments. Please find our response to all the comments below.
> 1. About low-cost initialization.
>      - How low-cost initializations are chosen.
> For the local search method used in our framework, a low-cost initialization is needed to realize its unique advantage in controlling the cost (Wu et al., 2020). We have made the low-cost initializations more clear as part of the input information in Section 3 and discussed how they are chosen in Section 4 of the updated paper.
>      - BS's performance without low-cost initialization.
> To better answer this question, we included an additional experiment in which random initialization is used for all methods (results included in Figure 11 in the appendix). This experiment shows that even without low-cost initialization, BS still performs the best (though with smaller margins).
>      - About GPEIPS with low-cost initializations.
> The reviewer has an idea of using a set of low-cost initial configs to improve GPEIPS. It is an interesting idea to try if one wants to design new methods based on GPEIPS. However, it requires investigation on how to construct the set (BS only takes one single initial point) and how large it should be.
> 2. About the interplay between global and local search.
> We thank the reviewer for asking this question and your intuition does reflect some part of the process. Our proposed scheme does leverage both global and local search. This claim can be supported from two perspectives. First, conceptually, it is indeed possible that global search may get rejected repeatedly at the beginning of the search. But as a local search thread expands its `territory' or converges, the admissible region is also expanded. It becomes less likely to repeatedly reject configs proposed by the global search. The admissible region will eventually cover the whole search space and the global search will not be rejected.  Second, the interplay between global search and local search can be easily observed in the case study shown in Figure 1(b). The diamonds correspond to BO-suggested trials in BS, as explained in the subcaption. This figure shows that the global search indeed plays a role after 1000s. The figure also shows that global search is not only responsible for directly finding better configurations, but also for creating local search threads that can achieve yet better performance.
> 3. Response to detailed questions.
>     - Details about the choice of $\Delta_i$ and the definition of reachable are added in Appendix A. How Continuous-integer-categorical hyperparameters are handled is added in Appendix B.
>     - About Trust region BO.
>     TurBO cannot handle categorical hyperparameters and it is not designed for typical ML model tuning. It is based on the specific BO method GP, and focused on the high dimensional robot, planning or controller problems with high resource consumption (50 to 100 workers in parallel), without taking care of heterogeneous cost and low resource requirement. Same as the question for GPEIPS, how to construct a beneficial set of low-cost initial points for TurBO and how large the set should be are not clear.
>     - We agree and thank your suggestion about the notation of the problem formulation. We have revised it accordingly to make the formulation more precise.
>     - About Figure 4(a) and 1(b).
>     Both figures convey the same message: BS has better anytime performance, and BO gets closer to BS with more time.
>     - About the definition of the speed of improvement.
>     The formula for the speed of improvement is only valid when there is at least one improvement. Otherwise, we do not have enough information to estimate the speed of improvement. We set the speed to the highest value in this initial phase due to an implicit assumption of diminishing return: the speed of improvement decreases as more resource is spent.
>     - About Eq. (2).
>     We appreciate your intuition and justification. S.s in Eq. (2) is the current speed, which is an upper bound of the future speed assuming the speed of improvement has a decreasing trend due to diminishing return. So we remove the negative term $S.c^{1st}-S.c$ from the formula you suggested to rectify the underestimation of the cost to reach $l$. To further improve over $l$, we assume it takes at least the same cost (again due to diminishing return), hence the factor 2 is added.
>     - Priority(S)'s connection to UCB.
>     We appreciate the reviewer's insightful comment. With a diminishing return assumption, Priority(S) is indeed an optimistic estimation on low the loss can become in the future. How far to look ahead in the future, i.e., the value of $b$ in Eq. (3) depends on how much resource is needed for every search thread to have a better performance than the currently best, and how much budget is left.
>     - Fidelity in local search. Please refer to the added discussion about fidelity in Section 5 of the updated paper.

---

### Official Review · AnonReviewer1 · 2020-10-27
**Official Blind Review #1**

**Rating:** 6
**Confidence:** 3

**Review:**

Summary:
This paper proposes BlendSearch, which combines global and local optimisation for the problem of hyperparameter optimisation when search cost is heterogenous. To achieve so, they use the combination of one global search instance (e.g. Bayesian optimisation; used to identify promising regions as starting points for local search) with multiple local search instances (which actually do the search). The local search instances will be created, merged and deleted on the fly using the criteria proposed by the authors. The paper finally experimentally validates their approach in various hyperparameter tuning experiments to show promising results.

============================================================================
Pros (in addition to the summary provided)
-	The paper addresses an important problem and proposes a sensible strategy that builds on two successful ingradients. The paper is also quite well-written and easy to follow.
-	A plus in my opinion (not necessarily emphasised by the authors) is that this framework seems to be a plug-in improvement that is agnostic to the exact global and local optimisation algorithms used. Therefore, it can be potentially be used in a wide range of setups.

============================================================================
Cons
There is generally no theoretical discussion in this paper. While the method can still be successful empirically, the reviewer thinks that having some sort of theoretical expositions will strengthen the paper and would more strongly motivate some of the algorithm design decisions the authors made. For example, the most important design is Step 1 of Blendsearch which is intended to balance exploration and exploitation, and it looks like the current approach (Eq (2)-(3)) are predicting the cost by the simple heuristic of extrapolating (linearly) past the observations, and I’m not sure how true this simplistic model is in reality. Given that a major selling point of this paper is about ‘cost awareness’, I feel that this aspect needs a bit more clarification/justification.

The idea of maintaining a simultaneous pool of different optimization instances is not new, and one example is TurBO-M that the authors cited, which formulate the selection of optimization instances as an implicit multi-armed bandit (MAB) problem.  While TurBO does not consider the cost heterogeneity and do not use local optimization instances, I think that their selection method is more principled and could be food for thoughts for the authors.

Another question is did authors consider batch version of BlendSearch, as currently the paper seems to rely on a purely sequential setup? Search in a parallel (and possibly asynchronous) way would greatly improve the applicability of the method in a practical setup. It would be nice to have discussions how easy it is to parallelize BlendSearch without degrading its performance.

Minor
-	In Fig 2, is the bottom leftmost diamond missing a ‘No’ path? I suppose there should be a No and a path straight from that diamond to upper rightmost ‘Search Thread Updater & Cleaner).

---

> ### Author Response · Authors · 2020-11-20
> **Response to  AnonReviewer1**
>
> We thank the reviewer for highlighting one of the appealing properties of our method. Our framework is indeed flexible enough to incorporate various global and local optimization algorithms, which makes it extensible when new BO or LS methods are invented.
> 1. Theoretical expositions.
> We thank the reviewer for this suggestion. We have updated the draft adding more theoretical expositions.
> Our search thread selection strategy follows the principle of optimism in the face of uncertainty from the multi-armed bandit literature to balance exploitation and exploration: We make an optimistic estimation on the future reward of each particular arm. Assuming each search thread has a diminishing return, the speed of improvement decreases as more resource is spent. Then the linear extrapolation provides a first-order upper bound of the future reward. Although the bound is not necessarily tight, it is refined adaptively and locally. It makes use of historical observations without imposing strong stochastic assumptions needed by UCB.
> Existing MAB solutions and their theoretical guarantees are not directly applicable to our novel problem setting. The goal of our problem is to minimize the best loss any single search thread can achieve using constrained resource. First, the loss of each search thread does not follow a stochastic distribution, while methods such as Thompson sampling (Agrawal and Goyal, 2012) or UCB (Auer et al., 2002) require that assumption to construct the covariance matrix of the posterior distribution or concentration bounds.
> Second, we have a goal of cost minimization and the cost of each trial is heterogeneous and unknown.  That makes classical methods for non-stochastic bandits inapplicable either. Third, the number of local search threads changes dynamically and the threads have interactions. Rigorous theoretical analysis for this novel MAB setting demands a series of theories. It is an interesting line of future work.
> 2. About batch version of BlendSearch.
> Batch version is not a primary goal of BlendSearch as BlendSearch is designed mainly for the low-resource setting. In the low-resource setting, it is more desirable to use the limited resource in one trial at a time instead of running multiple trials together. But it is easy to realize parallelization in BlendSearch, as discussed in the appendix. The design of having multiple independent local search threads naturally allows efficient asynchronous parallel trials and full utilization of the parallel resources. The design of utilizing existing global optimization methods allows existing easy-to-parallelize global optimization (such as random search or TurBO) to be plugged in. The prioritization of search threads is still useful as long as the number of search threads is larger than the maximal concurrent number of trials. Also, since our method can be used together with ASHA, it can naturally inherit the easy-to-parallel and asynchronous resource scheduling in ASHA when used in the parallel setting.
> 3. Minor: The leftmost diamond corresponds to line 6 in Alg 1. For both 'Yes' and 'No', the path from config evaluator to search thread updater & cleaner on the right (Line 9-11 in Alg 1) will be executed after the path on the left. We added numbered annotations in the figure to make that more clear.
>
> References:
>     - Shipra Agrawal, and Navin Goyal. "Analysis of thompson sampling for the multi-armed bandit problem." Conference on learning theory. 2012.
>     - Peter Auer, Nicolo Cesa-Bianchi, and Paul Fischer. "Finite-time analysis of the multiarmed bandit problem." Machine learning 47.2-3 (2002): 235-256.

---

### Author Response · Authors · 2020-11-20
**Paper revision summary**

We thank all the reviewers' constructive comments and suggestions on our work. We have revised our paper accordingly. The main changes are summarized as follows.

1. In Section 3, we revised the input specification and simplified the description of the admissible region.
2. We added Section 5, which includes the following content.
    - An intuitive theoretical justification of our method.
    - Details about how multi-fidelity pruning is used in our method.  And an additional experiment about multi-fidelity baselines on tree-based models.
    - A discussion about the role of low-cost initialization in our method, how it is chosen, and our method's robustness to it.
3. In Appendix A and Appendix B.1, we added detailed settings related to the local search method.
4. In Appendix B.2, we added an additional experiment for the hypothetical setting of not initializing the search with a low-cost point.

---

### Decision · Program_Chairs · 2021-01-07
**Final Decision**

**Decision:**

Accept (Poster)

**Comment:**

This paper presents a hyperparameter optimization (HPO) method in which two search strategies: global and local optimizations, are effectively combined. All reviewers evaluated the proposed method positively. The experimental results clearly show the effectiveness of the proposed method, and it could be an important contribution to the AutoML research community. On the other hand, since there is no theoretical justification for the proposed method, it is not clear why the performance of the proposed method is improved so much. The author's rebuttal has alleviated some of our concerns on this point, but the further theoretical analysis is desirable.